# The Use of Sand Substrate Modulates Dominance Behaviour and Brain Gene Expression in a Flatfish Species

**DOI:** 10.3390/ani13060978

**Published:** 2023-03-08

**Authors:** Maria Mafalda Almeida, Elsa Cabrita, Elvira Fatsini

**Affiliations:** Centre of Marine Sciences (CCMAR), Faculty of Science and Technology, University of Algarve (Campus Gambelas), 8005-139 Faro, Portugal

**Keywords:** social category, physical enrichment, fish welfare, Senegalese sole, aquaculture management

## Abstract

**Simple Summary:**

Adding physical enrichment to the rearing conditions may promote fish welfare and reduce detrimental characteristics that fish develop in captivity. Senegalese sole (*Solea senegalensis*), an important species for European aquaculture, is reared in intensive conditions using fibreglass tanks. This species shows reproductive dysfunctions that do not allow it to complete its life cycle in production. Recently, dominance behaviour has been studied to try to solve this problem. The present study aimed to assess the effect of sand as an environmental enrichment in the dominance behaviour and brain gene expression of Senegalese sole juveniles. Several behaviours associated with feeding and territoriality were analysed by video recordings. In both environments, dominant soles were the first to feed, displayed a higher number of contact interactions and dominated the area close to the feeding point, where the events were reduced in fish maintained in the sand. The *nr4a2* and *fshra* genes related to differentiation of dopamine neurons and regulation of maturation were significantly up-regulated in dominant fish maintained with sand compared to dominants maintained without sand. Using an enriched environment may affect Senegalese sole dominance, enhance welfare and advance future maturation.

**Abstract:**

Physical complexity adds physical enrichment to rearing conditions. This enrichment promotes fish welfare and reduces detrimental characteristics that fish develop in captivity. Senegalese sole (*Solea senegalensis*) is an important species for European aquaculture, where it is reared in intensive conditions using fibreglass tanks. However, reproductive dysfunctions present in this species do not allow it to complete its life cycle in captivity. Recently, dominance behaviour has been studied to try to solve this problem. The present study aimed to assess the effect of sand as environmental enrichment in the dominance behaviour and brain mRNA abundance of Senegalese sole juveniles. Four tanks of sole (*n* = 48 fish in total) were established in two different environments (with and without sand). Juveniles were subjected to dominance tests of feeding and territoriality. Behaviours analysed by video recordings related to the distance from the food delivered and harassment behaviour towards other individuals (e.g., resting of the head on another individual). In both environments, dominant sole were the first to feed, displayed more head-resting behaviour and dominated the area close to the feeding point, where the events were reduced in fish maintained in the sand. mRNA expression related to differentiation of dopamine neurons (*nr4a2*) and regulation of maturation (*fshra*) were significantly upregulated in dominant fish in the sand environment compared to dominants maintained without sand. The use of an enriched environment may affect Senegalese sole dominance, enhance welfare and possibly advance future maturation.

## 1. Introduction

Aquaculture is one of the fastest-developing growth sectors, becoming an important economic activity in many countries. Nevertheless, the aquaculture sector is still far from tackling all the issues to optimize its high potential [1]. Some of these problems are associated with the final adaptation to captive conditions of some fish species which are considered valuable for the aquaculture industry. For this purpose, it is very important to comprehend the ecology of fish species with domestication problems in aquaculture production.

Fish in captivity might develop maladaptive or undesirable traits, which may lead to rapid selection for domestic genotypes, sometimes within a single generation [2,3]. Environmental enrichment is a strategy to avoid these undesired traits, especially when fish are cultured for restocking purposes or when fish are used as model organisms in laboratories. This strategy is a method to increase environmental complexity by introducing physical items into the rearing environment (e.g., plants, objects, substrates) and is believed to be one of the most valuable and promising methods to enhance fish welfare [4,5] in aquaculture production. Environmental enrichment was also demonstrated to enhance fish growth rate [6] and neural plasticity [5], decrease aggressive and anxiety-like behaviour [7], reduce maintenance metabolism and promote survival-related behaviours of released fish (e.g., behavioural flexibility) [8]. One environmental enrichment is the use of tank floor substrates which have several beneficial effects for fish that interact with the bottom, like flatfish species. These substrates might reduce injuries in fish that usually rest at the bottom of the tank [2]. For example, sand eliminates outbreaks of black patch necrosis on the common sole (*Solea solea*), since sand removes pathogen-promoting dead cells from the fish bodies. Tank substrates provide an opportunity to display burying behaviours in benthic species [9,10]. Moreover, sand reduces respiration and resting metabolic rates in the common sole, showing that this substrate provides a less stressful environment [11].

Additionally, environmental enrichment has also been observed to modulate social behaviour, above all reducing aggressiveness and gene expression in several fish species, including zebrafish (*Danio rerio*) [12], black rockfish (*Sebastes schlegelii*) [4], sea bream (*Sparus aurata*) [13,14] and Nile tilapia (*Oreochromis niloticus*) [8], among others. Dominance is considered a social behaviour defined as success in competition over limited resources such as food, specific areas, shelter, mates, spawning areas and offspring [15]. Usually, dominant animals have more access to food and shelter and higher mating success, and they are less likely to be predated, whereas subordinate animals have lower reproductive success, chronic stress and reduced disease resistance [16,17]. In fish, social hierarchies are often categorized by behaviours that are habitually registered through feeding contests or territoriality [18]. In this context, environmental enrichment might help to ensure proper ethological conduct and improve the welfare of farmed fish.

Senegalese sole is an emerging flatfish species for aquaculture because of declining market supply and high market value. Furthermore, this species has good characteristics for cultivation, such as high growth and survival rates, and a high capacity to adjust to intensive production [19]. However, this species faces some bottlenecks regarding its production. The main problem is reproductive dysfunction in cultured male breeders that do not fertilize the eggs released by females [20,21]. This forces the use of wild-origin breeders, which can spawn spontaneously in captivity, but this is unsustainable in the long term. Moreover, this species used to be reared in intensive conditions through recirculation aquaculture systems (RAS) using fibreglass tanks [19]. The study of dominance in this species under production conditions has been one approach to understanding reproductive dysfunction. There is a clear hierarchy in Senegalese sole populations in captive conditions, particularly during reproduction. Only 6 to 48% of the individuals in a population contribute to reproduction [21,22]. However, the features of this hierarchy and dominance remain unknown. Recently, dominance categories (dominant and subordinates) for Senegalese sole juveniles related to feeding behaviour and territoriality were reported [17]. The knowledge of dominance behavioural patterns in this species would be beneficial for stock management in aquaculture since failed spontaneous spawning might be related to social interactions and dominance. Additionally, studies in other fish species found that social behaviour might be linked to differential gene expression since behavioural phenotype differences are associated with different levels of biological regulation measured in the transcriptome [23]. Transcripts such as brain-derived neurotrophic factor (*bdnf*) and nuclear receptor subfamily 4, Group A, member 2 (*nr4a2*) were also associated with social status in zebrafish [24], as these biomarkers are implicated in neuroplasticity [25] and in the preservation and differentiation of dopaminergic neurons [26], respectively. The heat shock protein 90, alpha (cytosolic) class A (*hsp90aa*) and the heat shock protein 70 (*hsp70*) were both related to cellular stress and cellular protection [27] in Senegalese sole. The target gene glyceraldehyde-3-phosphate dehydrogenase isoform 2 (*gapdh-2*), which is involved in metabolic pathways [28], has been observed to be differentially expressed depending on behavioural traits [29,30].

The objective of this study was to assess the dominance categories of Senegalese sole juveniles related to the presence of a substrate by observing specific registered behaviours associated with feeding and territoriality and the expression of some selected mRNA transcripts related to social behaviour and maturation.

## 2. Materials and Methods

### 2.1. Animal Rearing Conditions

A total of 48 cultured Senegalese sole juveniles (203.25 ± 14.51 g), 1 year and 8 months of age, were used for the present study. The specimens were maintained at Ramalhete station (University of Algarve, Faro, Portugal) and were randomly distributed among 4 different white plastic tanks (52 cm height, 57 cm width and 97 cm length: 95.8 L of water) such that each tank contained 12 fish. The 4 tanks were exposed to artificial light simulating a natural photoperiod for the winter season (10L:14D) and were integrated into an open flow system, meaning that the temperature of the water varied accordingly to the temperature of the Ria Formosa (37°00′22.2″ N 7°58′02.3″ W/11–15 °C in the winter). Water parameters (temperature, oxygen and salinity) were monitored daily to ensure optimum rearing conditions. Fish were fed once a day ad libitum (approximately 2% of biomass [19]) using 4 mm commercial pellets from SPAROS Lda (Olhão, Portugal) (marine sole) on a daily basis.

### 2.2. Behavioural Assay

#### 2.2.1. Experimental Dominance Set-Up

For this study, a specific structure was built to cover the 4 tanks from the sides and the front, with shade nets, to avoid direct light contact due to Senegalese sole sensitivity to light and to prevent operational influence on behaviour. A feeding tube (57 cm PVC tube) was placed in the front right corner of each tank, and one colour camera (MVC 212 WP—LED, China) was introduced above to give a view of the entire tank (Figure 1A,B). The digital cameras were used to record the dominance test behaviour of the Senegalese sole juveniles. The cameras were connected to a digital video recorder (16CH Digital Video Recorder Case, OYN-X by CCTV) programmed to record from 6.00 am to 6.00 pm. The cameras were recording throughout the whole experiment, monitoring remotely with the software General CMS, commonly used for H.264 DVR format.

#### 2.2.2. Dominance Test

All Senegalese sole were tagged at the moment of their arrival at Ramalhete station using a passive integrated transponder (PIT) tag (ID100 Implantable Transponder, Trovan, Aalten, The Netherlands). For this purpose, sole were anaesthetized by immersion with 2-phenoxyethanol (Merck, Lisbon, Portugal) and randomly distributed in the different tanks. At the same time, Senegalese sole were individually photographed giving a number to each fish and associated with the PIT-tag number. The images were focused on the fish shape and caudal fin structure (Appendix A). All fish were weighed at the beginning and the end of the experiment to calculate the growth parameters to correlate with the social category. Less than one centimetre of sand from the Ria Formosa was added to two tanks as an environmental enrichment to mimic natural conditions, resulting in duplicates of tanks with and without sand. This quantity of sand was added to allow observation and distinguish the individuals. Fish were acclimated to the new treatment for 5 days. After this period, all fish were acclimated to the food delivered through the feeding tube always at the same time (10.00 am according to operational routines) for 3 days. Before the experiment started, the fish were fasted for 48 h to equalize individual conditions (Figure 2A).

The dominance test was performed on 4 consecutive days where Senegalese sole were fed approximately 1.5% biomass, i.e., less than the usual ration provided, to promote feeding motivation. Food was delivered directly through the feeding tube. This setup triggers territoriality and feeding competition in this species, where dominant fish tend to monopolize the feed delivery point [17].

Dominance categories (dominant, intermediate and subordinate) were determined according to feeding order and associated with the events of each behaviour registered. This categorization was observed for the 4 consecutive days of the dominance test to verify the consistency in social category classification for the 4 days. Thus, the first 3 animals to eat were considered dominant, and the last 3 fish to eat were considered subordinate in the same tank, while the other fish were considered intermediate. Behaviours were recorded and registered according to Fatsini et al. [17] and described in Table 1. Briefly, Feeding Order (FO) behaviour was defined as the order in which a fish ate and was the parameter used to classify the fish as dominant, intermediate or subordinate as described above. The positions of each fish (before and after feeding) were classified into 6 positions (zones) relative to the feeding point. These parameters are associated with territorial interactions. The fish located in the zone closest to the feeding point was ranked 1, second closest was ranked 2, etc., until rank 6 (Figure 2B). To obtain the frequency of Position Before Feeding (POSITB), events were observed by analysing the videos 30 min before the food was delivered every 5 min (6 frames in total). For Position After Feeding (POSITA), videos were analysed in the same manner from the beginning of food delivery until the last pellet was eaten. To analyse the positions in the video recordings, the zones were drawn in a template to overlap in the images using the software editor VEGAS Pro 17.0. In the case that one fish was located between two zones, the zone considered was where the head was positioned. Resting the Head (RTH) was a behaviour related to social interaction among the individuals for which the frequency was analysed, noting the events before and after the food was delivered every 5 min according to the experimental feeding period per treatment (more details in Table 1).

At the end of the experimental period, after dominance classification, each tank was sampled where 3 dominant and 3 subordinate fish corroborated with the PIT-tag number were euthanatized by anaesthetic overdose (1000 ppm 2-phenoxyethanol, Sigma-Aldrich, Schnelldorf, Germany). Individual biometric data (weight, length and sex) were registered, and the brain was dissected, placed in liquid nitrogen and stored at −80 °C for further molecular analysis.

### 2.3. RNA Isolation, Complementary DNA Synthesis and Quantitative Real-Time Polymerase Chain Reaction Assay

The expression of brain target transcripts (*c-fos*, *nr4a2*, *nrd2*, *bdnf*, *hsp90aa*, *hsp70*, *gapdh*, *pgr* and *fshra*) (Table 2) for dominance behaviour was measured in the brain of 24 fish (12 dominant and 12 subordinates). Target genes were chosen according to their biological significance (cognitive capacity (*c-fos*, *nr4a*, *nrd2* and *bdnf*), stress response (*hsp90aa* and *hsp70*), basic metabolism (*gapdh*) and maturity stage (*prg* and *fshra*)) (Table 2) [17,30].

RNA was extracted from Senegalese sole brains using TRI Reagent RNA Isolation (Merck, Lisbon, Portugal) following the manufacturer’s instructions. RNA quantity and purity were determined using a Nanodrop spectrometer (Nanodrop 1000, Thermo Scientific, Porto Salvo, Portugal). Afterwards, RNA was purified using NucleoSpin RNA Plus Kit (Macherey-Nagel, Düren, Germany), performing a rDNase digestion to remove genomic DNA. The isolated RNA showed high purity (A260/280 > 1.9) and was stored at −80 °C until further use.

Complementary DNA (cDNA) was synthesized from 1 µg RNA using the cDNA synthesis kit (Thermo Scientific Maxima First Strand cDNA Synthesis Kit, Thermo Scientific, Porto Salvo, Portugal) following the manufacturer’s protocol performing a second dsDNase digestion to remove possible traces of genomic DNA which may interfere with qPCR results. Reverse transcription (RT-PCR) conditions were 25 °C for 10 min, 50 °C for 30 min and 85 °C for 5 min.

Before performing the real-time PCR, primers were validated by conventional PCR using 2 pools of cDNA separated by sex (male or female). The GoTaq G2 Flexi DNA Polymerase (Promega, Carnaxide, Portugal) was used to perform conventional PCR with the following conditions: initial denaturation step at 98 °C for 1 min, followed by 35 cycles of denaturation at 94 °C for 30 s, annealing at Tm (57–63 °C) for 30 s and extension at 72 °C for 1 min. Afterwards, the amplified product was loaded in an agarose gel (1.5%) in order to observe the PCR amplification of the transcript products.

Primer efficiency was evaluated by serial dilutions, where primers less than 85% of efficiency were discarded. The qPCR was run using a Bio-Rad CFX96™ Thermocycler (Bio-Rad, Amadora, Portugal) in 96-well plates in duplicate. Reactions were performed in 20 µL volumes containing 10 µL of SsoFast EvaGreen Supermix (Bio-Rad, Amadora, Portugal), 2 µL of the primers (0.5 mM) (Table 2) and 5 µL of cDNA at the validated dilution. Furthermore, amplifications were carried out with a systematic negative control (NTC: no template control) containing no cDNA. Standard amplification conditions contained an initial denaturation step at 95 °C for 30 s, followed by 40 cycles of denaturation at 95 °C for 5 s, annealing at 55 °C for 5 s and a melt curve with a 0.5 °C increase (65–95 °C) for 2–5 s.

Results were normalized using two housekeeping genes, ubiquitin (*ubq*) and beta-actin (*β-actin*), calculating the geometric average [33]. The Pfaffl method using modified 2ΔΔ [34] was applied to obtain the mRNA abundance for each gene. The treatment without sand was used as the control group, since it is the usual condition to maintain this flatfish species in captivity.

### 2.4. Statistical Analysis

The statistical analysis was performed with SPSS Statistics 25.0 software (IBM Co., Hong Kong) and plotted with GraphPad Prism 9 software (GraphPad Software, Inc., Boston, MA, USA). All results are presented with means ± standard deviation (mean ± SD). The Shapiro–Wilk test was used to analyse data normality, and logarithmic transformation was performed when data were not normally distributed. For all cases, the level of significance was considered with a *p*-value < 0.05.

For behavioural analysis, the consistency of the results was verified for the variable Feeding Order (FO) of each individual by applying RM-ANOVA. The same operator watched the video recordings to avoid bias in the results. Since there was consistency over the 4 days in the dominance category, the mean of each behaviour was calculated. Mixed-effect nested ANOVA was used to observe differences among dominance categories (dominant, intermediate and subordinate), where the treatment effect appears nested within the tank as a random factor and the dominance category as a fixed factor. A general linear model (GLM) was carried out to analyse the growth parameters during the experiment associated with social categories. For the qPCR, a two-way ANOVA was used to assess the differences between dominant and subordinate fish from different treatments, with and without sand. Raw data from behavioural analysis and qPCR expression are available in *figshare* (https://doi.org/10.6084/m9.figshare.16910380.v1 (accessed on 31 October 2021)).

## 3. Results

Senegalese sole behaved differently when exposed to an enriched environment with sand. The fish in the tanks without sand were more active in terms of feeding behaviour, actively looking for the food delivered in the tank. The fish in these tanks approached the feeding tube and crowded the area before the food was delivered. As a result, these fish finished eating all the food by 10.30 am. This behavioural pattern was not observed in the tanks with the sand environment. Fish in the enriched environment barely moved for long periods, since most of them were buried in the sand. When the time of feeding approached (10.00 am), the fish remained quiet for at least 30 min before starting to swim towards the feeding tube when the food was already in the tank. This meant that the sole in these tanks only finished eating all the food at noon.

### 3.1. Dominance Test

There was consistency in the Feeding Order (FO) of Senegalese sole individuals throughout the 4 days of the dominance test (*p* = 0.678). According to this feeding rank, Senegalese sole were classified as dominant, intermediate or subordinate. Therefore, 6 fish were classified as dominants, 6 as subordinates and 12 as intermediates per treatment, with and without sand. No tank effect interaction was observed, neither in treatment (sand substrate) nor in the social category, on any of the parameters evaluated (POSITB: *p* = 0.053; POSITA: *p* = 0.859; RTH: *p* = 0.810).

The FO was observed to be directly connected with the position of the individuals. The POSITB was significantly different among categories in Senegalese sole kept without sand. Dominant fish occupied positions near the feeding tube most in comparison to subordinates (*p* = 0.013; Figure 3). However, this parameter was similar in the intermediate sole (*p* = 0.463). No differences were observed between intermediates and subordinates (*p* = 0.061; Figure 3). Significant differences were noted for POSITB among categories of fish reared in the sand, where dominant sole occupied positions nearer the feeding tube than intermediates (*p* = 0.045; Figure 3) and subordinates (*p* = 0.007; Figure 3); however, intermediate fish occupied similar positions to subordinate ones (*p* = 0.412; Figure 3). There were no differences between treatments. The behavioural pattern of POSITB behaviour was similar in fish maintained without and with sand. However, there was more variability in fish maintained in the sand in all social categories due to the low swimming activity of these individuals.

POSITA was significantly different among categories for fish maintained without sand. Dominant fish occupied positions nearer to the feeding tube than subordinates (*p* = 0.002; Figure 4) and intermediates (*p* = 0.031). However, intermediate fish occupied similar positions to subordinate fish (*p* = 0.157; Figure 4). For Senegalese sole reared in the sand, the dominant individuals also occupied positions nearer to the feeding tube than the subordinates (*p* = 0.007). However, intermediates occupied similar positions to dominants (*p* = 0.377) and different from subordinates (*p* = 0.031; Figure 4). Observing both treatments, Senegalese sole reared without sand occupied positions significantly closer after (POSITA) the food was delivered in all social categories than animals kept in the sand. The behavioural frequency of POSITA behaviour for fish maintained in the sand was based on the delay observed in this environment, where the animals started to feed 30 min after the food was delivered and were delayed 2 h in eating all the food present in the tank, whereas the animals maintained without sand ate their food within 30 min after delivery.

The RTH index showed significant differences among categories for Senegalese sole reared without sand. Dominant fish rested the head on other fish more frequently than subordinates (*p* = 0.005; Figure 5), which received more head rests from other fish. Intermediate fish displayed this behaviour similar to dominants and subordinates (*p* = 0.768; Figure 5). In fish kept in the sand, there were significant differences among categories. Specifically, the dominant fish behaved differently than the subordinates (*p* = 0.006; Figure 5) but did not behave differently than the intermediates (*p* = 0.897; Figure 5). However, intermediates behaved differently than the subordinates (*p* = 0.0001; Figure 5).

There were no significant differences due to the high variability of the data from the sand treatment. However, the behavioural pattern of RTH behaviour for fish from both environments was similar, whereby dominant fish rested the head more frequently on other fish and subordinate fish received more head rests from other individuals. The behavioural frequency of fish maintained in the sand was based on the delay observed in this environment, where these animals had a low swimming activity; as a consequence, social interaction was more variable than in animals reared without sand.

In terms of size, there was no difference between social categories and growth parameters during the experimental period. Therefore, no significant differences between the weight of social status categories were observed, neither in fish kept without sand (dominant: *p* = 0.473; intermediate: *p* = 0.327; subordinate: *p* = 0.881), nor in fish maintained in the sand (dominant: *p* = 0.917; intermediate: *p* = 0.969; subordinate: *p* = 0.991; Table 3).

### 3.2. Gene Expression Analysis in the Brain

The expression of the nine transcripts tested was not significantly different between dominant and subordinate fish within the same treatment, which means comparing dominant vs. subordinate fish reared with or without sand.

Nevertheless, *nr4a2* (*p* = 0.049; Figure 6A) and *fshra* (*p* = 0.008; Figure 6B) were differentially expressed between dominant fish from both treatments. The two transcripts, *nr4a2* and *fshra*, showed the same expression pattern in dominant fish, i.e., down-regulated in dominant individuals maintained without sand and up-regulated in dominant individuals with sand. No differences were observed between subordinate fish from both treatments (Figure 6).

## 4. Discussion

### 4.1. Dominance Test

Flatfish species, like Senegalese sole, depend on visual and olfactory cues to search for food [35]. Previous studies showed that cultured fish demonstrate a tactic according to the food delivery, due to the domestication process [36]. This tactic varies according to the species ethogram achieving different strategies [36]. In the present study, this tactic was corroborated with the behaviours associated with territoriality, i.e., Position Before Feeding (POSITB) and Position After Feeding (POSITA), which proved that sole which were closer to the food delivery tube were the first to eat and were thus considered dominant. The consistency of this feeding behaviour among categories classified by the Feeding Order during the 4 days could be associated with the similarity of this behaviour to the pecking order observed in poultry which is established at a young age and remains stable for life [37]. Resting the Head behaviour is a usual behaviour, previously described in the ethogram of Senegalese sole [38]. Depending on the context, this behaviour could have several meanings; for example, males display this behaviour during reproduction to achieve direct contact with females to initiate the courtship [21,39]. In terms of dominance, this behaviour could be considered harassment due to, in several cases, a displacement behaviour being implicit by the direct contact performed among individuals [17]. This behaviour is very important in non-aggressive species like Senegalese sole, which need to display these types of behaviours to show the hierarchy within the population. Aggression during feeding behaviour is the most common method to establish hierarchies and, in several species, the dominant fish are often larger than the subordinate ones [39]. This study observed no differences between gaining weight and being dominant. Hence, the dominance behaviours in territoriality and social interaction performed by the different categories in both environments were not linked to animal size. These results agree with a previous study developed by our group [17], where dominance was suggested as a contributing factor to growth variation in this species, but alone was not responsible for the variations reported. However, it supported that other factors such as life strategy, appetite and feeding behaviour need to be considered.

Currently, there is an increasing number of studies oriented toward improving the habitat of cultured fish to ensure the welfare of aquaculture [4,8,14]. For example, Arechavala-López et al. [14] showed that gilthead seabream in the presence of plant-fibre ropes as an enriched environment in the tank increased antioxidant activity in the whole brain and dopaminergic activity, and they presented higher exploratory behaviour, spatial orientation and learning capability compared with those fish without enrichment. Zhang et al. [4] demonstrated that black rockfish in the presence of plastic plants decreased aggressive behaviour, stress response, and brain expression levels of several genes involved in neurogenesis. In the present study, a sandy substrate was used as an enriched environment to analyse the dominance behaviour in Senegalese sole juveniles. The results suggest that dominance behaviour was modulated in the presence of sand. Indeed, the behavioural pattern for all the parameters registered in fish from both treatments was similar, where the dominant sole displayed the behaviours analysed more than the subordinate fish. However, the frequency of these behaviours was lower in fish reared in the sand than in sole maintained without sand. This could be explained by the fact that animals kept in the sand showed a delay in feeding behaviour, just starting to eat 30 min after the food was provided, being buried in the sand and remaining quiet. These results showed the preference of sole to be buried in the sand rather than swimming directly to the food provided. However, the fish which ate first were also located closer to the feeding tube and considered dominant. Fatsini et al. [17] demonstrated that dominant sole, which ate first in a dyadic feeding response test, were also buried in the sand for more extended periods at the end of the place-preference test, showing the importance of territoriality in this species. Flatfish species display burying behaviour for several reasons, such as avoiding predation, ambushing prey and conserving energy by reducing activity levels [40]. Moreover, the burial strategy has positive consequences, like better skin condition [41], improved survival and growth and lower metabolic expenditure [11]. This has been associated with the fact that the sand may act as a shelter where fish spend less energy in moving around, and therefore require less food. Therefore, the substrate might be considered a key factor for sole welfare in aquaculture. It would be interesting to register the zonation regar40ding the sand in future dominance analysis in group tests to observe the capacity of the dominant sole dominating sand and the tube feeding. Regarding reproduction, Fatsini et al. [21] showed that sole breeders select tank areas containing sand to perform courtship, demonstrating a positive impact on reproductive behaviour. This conduct mimics the behaviours usually observed in nature for this flatfish species since sole has been described as sedentary [40]. However, some disadvantages have been described using the substrate as a physical enrichment in captive conditions, including that it can cause high organic loading, is time-consuming to clean, makes it hard to remove certain parasitic diseases and may alter water chemistry and quality [3]. In the present study, a small quantity of sand was used to facilitate the observation of the fish and to perform the maintenance routines. No mortalities or signs of sickness were observed during the period of the experiment, and the parameters of the water were registered every day without noticing changes between treatments. This has been observed in another study performed by our group which reared Senegalese sole juveniles in a sand environment for two years where the sand was changed partially every two months, and the routines were the same as the tanks without sand (unpublished data).

The use of sand as an environmental enrichment would increase the welfare of the sole by mimicking the natural conditions and saving energy, thereby improving its fitness. Moreover, it might act as a trigger for future reproductive performance.

### 4.2. Gene Expression in Association with Dominance Behaviour and Enriched Environment

The present study investigated some of the target genes involved in several functions (cognitive capacity, stress modulation, basic metabolism and maturation). In our study, none of the transcripts analysed (*nr4a2*, *c-fos*, *nrd2*, *bdnf*, *hsp90aa*, *hsp70*, *gapdh-2*, *pgr*, *fshra*) was differentially expressed between dominant and subordinate sole in both environments, with or without sand. However, Fatsini et al. [17] reported differences between dominant and subordinate fish in *c-fos* and *nrd2*, but only using tanks without enrichment. In the present study, genes associated with the learning process and neural plasticity (*nrd2* and *bdnf*) were similarly expressed in fish of different social categories or treatments. This result is contrary to several studies which reported different expressions from several fish species (Atlantic salmon (*Salmo salar*), gilthead seabream (*Sparus aurata*), among others) which were submitted to an enriched environment [5,14,42]. However, in those studies, the population was not divided into social categories. Regarding the stress modulation genes, *hsp90aa*, *hsp70* and *gapdh-2*, no differences were observed among social categories or sand enrichment. This was unexpected given that stress was decreased in the presence of environmental enrichment as observed in other fish species [5,14]. However, Senegalese sole showed similar expression of the *hsp90aa* gene when different tests were applied to classify juveniles in different stress-coping styles [30], showing that some stressful situations in Senegalese sole are not reflected in the heat shock protein gene expression. However, the same study [30], observed different expression of the *gapdh-2* gene among stress-coping categories. This gene is important in terms of basal metabolism and is associated with acute stress [28], showing that the presence or absence of sand would not affect acute stress in Senegalese sole. Comparing both environmental treatments, two transcripts exhibited significant differences in mRNA abundance between dominant sole from both environments. The transcript *nr4a2*, a gene encoding the protein NR4A2, which is a receptor associated with the differentiation of dopamine neurons [26], was significantly up-regulated in dominant fish maintained in sand compared with dominant individuals reared without sand. The imbalance of this gene has been associated with several neurodevelopmental disorders in mammals, such as Parkinson’s and epilepsy [43], showing the importance of the equilibrium of the expression of this gene in some specific parts of the brain. In addition, the activity and modification of dopaminergic neurons have been associated with behavioural patterns in several fish species such as medaka (*Oryzias latipes)* [44], midshipman (*Porichthys notatus*) [45] and a cichlid fish (*Astatotilapia burtoni*) [26]. This gene has been linked to behaviour, such as locomotor activity, at the early stages after hatching [46]. One study in zebrafish [46] and another in mice [47] showed that a lack of expression of this gene provoked an increase in locomotor activity. This change in behaviour in early stages can remain until adulthood and is more prominent in early stages than in adults [46,47]. This could be an explanation for why in our study, dominant fish without sand showed *nr4a2* down-regulation since this social category has been associated with higher activity.

Studies have shown that rearing Senegalese sole using an enriched environment might improve survival and growth and increase maturation. In our study, the dominant individuals from sand presented a higher expression of *fshra* than dominant individuals from tanks without sand. This fact might be associated with the presence of the sand, which may trigger the expression of the Fsh receptors in the brain which could be linked with possible advanced maturity. Furthermore, the maturation stage might be related to stress conditions, where it has been demonstrated that maturation might be affected by the stress condition. In this case, some fish species reared in substrates used as an environmental enrichment showed lower cortisol and decreased aggressive behaviour [4,7]. This transcript has a crucial role in supporting long-term gametogenesis in Senegalese sole [31,32]. An enriched environment might offset and stop the behavioural and neuroendocrine effects of stress by improving stress resilience.

The use of environmental enrichment deserves to be more investigated, particularly in fish reared under captive conditions, since recent studies have shown the implication of this enrichment in intergenerational behaviour and epigenetic effects on fish. Further research regarding gene expression and enriched environment is needed to better understand the implication of this relationship in Senegalese sole.

## 5. Conclusions

In this study, different behavioural frequencies were observed when Senegalese sole were exposed to an environment with sand substrate. Such an environment was demonstrated to be helpful to the welfare of this fish species by reducing dominant behaviour and possibly enhancing future maturation. Moreover, this study showed two genes (*nr4a2* and *fshra*) differentially expressed between dominant sole from both environments. In the case of Senegalese sole, these results showed the importance of preserving natural conditions to modulate behaviour and reduce phenotypical variation in this species, above all when they may be associated with reproduction. However, more research is needed to link hierarchical distribution in juveniles to breeders and use this methodology to ensure future reproductive success.

## Figures and Tables

**Figure 1 animals-13-00978-f001:**
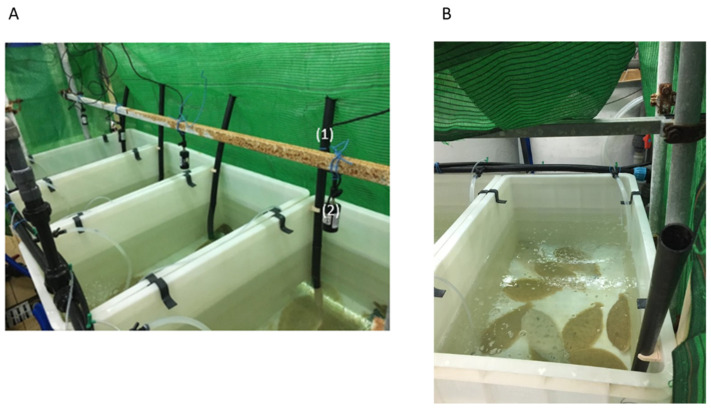
Experimental structure to conduct dominance test in Senegalese sole juveniles. (**A**) View from the inside with the shade cloth down. (1) Feeding tube and (2) Digital camera. (**B**) View from the front of the tank; note the feeding tube.

**Figure 2 animals-13-00978-f002:**
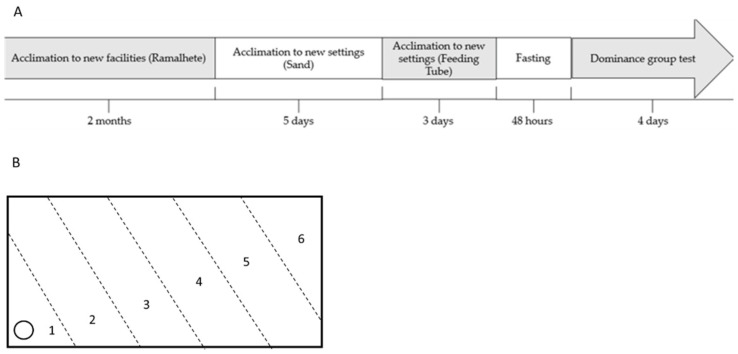
(**A**) Chronogram of the dominance experimental design in Senegalese sole juveniles. (**B**) Scheme of the areas of the different positions used to rank the fish for “Position Before Feeding” (POSITB) and “Position After Feeding” (POSITA) behaviours in the dominance test. Different position areas (1–6) are shown by dashed lines. Circle: The feeding tube consisted of a PVC tube to deliver the food (adapted from Fatsini et al. [17]).

**Figure 3 animals-13-00978-f003:**
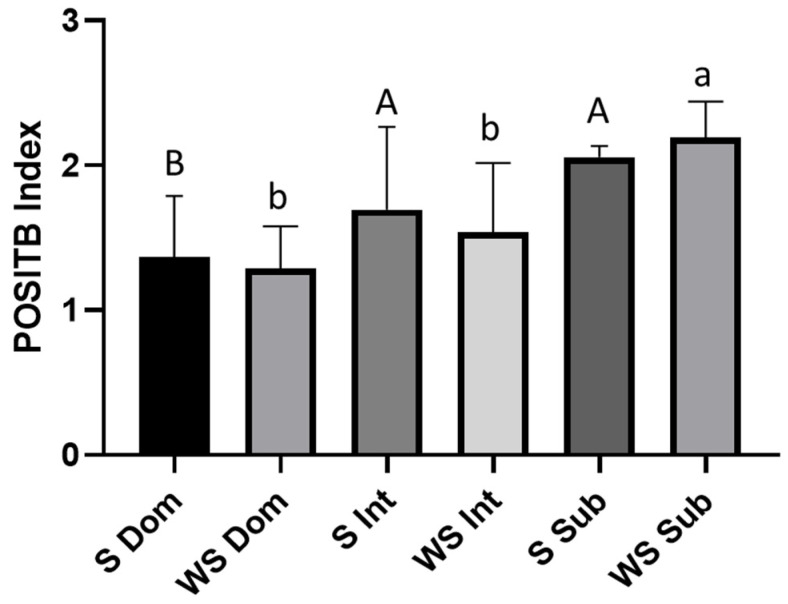
Positions before feeding (POSITB index) in Senegalese sole juveniles for dominant (*n* = 6), intermediate (*n* = 12) and subordinate fish (*n* = 6) per environment. S = Sand; WS = Without Sand; Dom = dominant; Int = Intermediate; Sub = Subordinate. Data are shown as mean ± SD. Different letters indicate significant differences within the same environment (two-way ANOVA; *p* < 0.05).

**Figure 4 animals-13-00978-f004:**
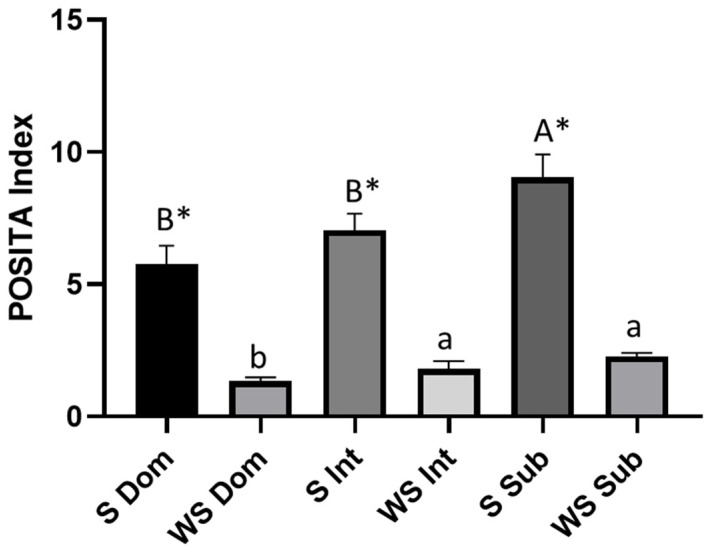
Positions after feeding (POSITA index) in Senegalese sole juveniles for dominant (*n* = 6), intermediate (*n* = 12) and subordinate fish (*n* = 6) per environment. S = Sand; WS = Without Sand; Dom = dominant; Int = Intermediate; Sub = Subordinate. Data are shown as mean ± SD. Different letters indicate significant differences within the same environment; (*) denotes significant differences between treatments within the same social category (Two-Way ANOVA; *p* < 0.05).

**Figure 5 animals-13-00978-f005:**
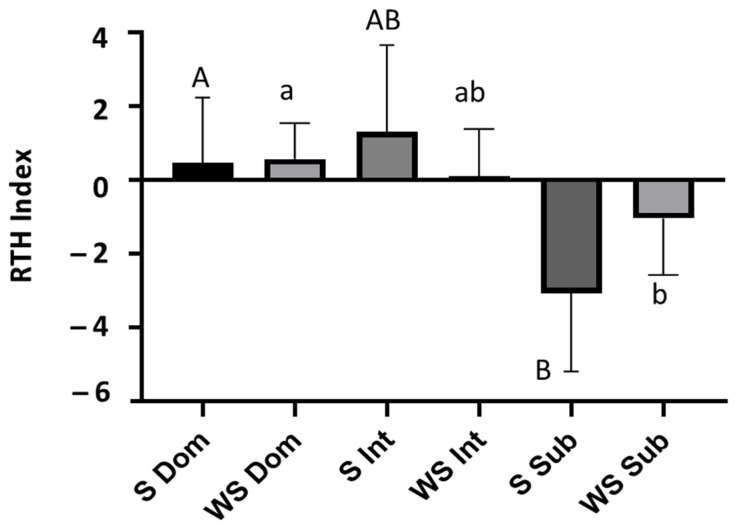
Rest the Head (RTH index) in Senegalese sole juveniles for dominant (*n* = 6), intermediate (*n* = 12) and subordinate fish (*n* = 6) per environment. S = Sand; WS = Without Sand; Dom = dominant; Int = Intermediate; Sub = Subordinate. Data are shown as mean ± SD. Different letters indicate significant differences within the same environment (two-way ANOVA; *p* < 0.05).

**Figure 6 animals-13-00978-f006:**
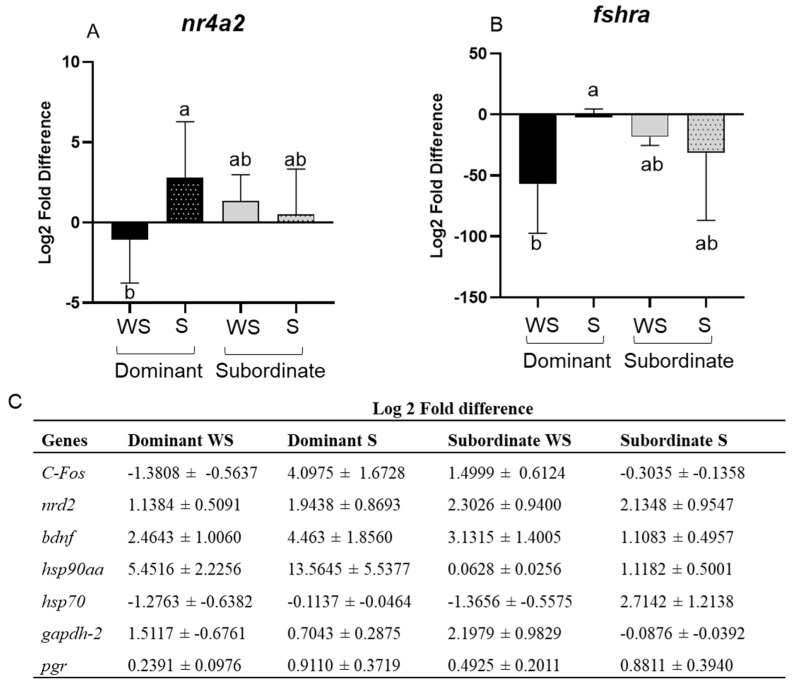
Brain mRNA abundance of Senegalese sole juveniles from different dominance categories in both environments, dominant without sand (WS) (*n* = 6) and with sand (S) (*n* = 6) and subordinate without sand (WS) (*n* = 6) and with sand (S) (*n* = 6). (**A**) *nr4a2*, (**B**) *fshra*, and (**C**) Table showing the expression of the other examined transcripts (*c-fos*, *nrd2*, *bdnf*, *hsp90aa*, *hsp70*, *gapdh-2* and *pgr*). Data are shown as means ± SEM. Different letters indicate different expression among social categories (Two-Way ANOVA; *p* < 0.05).

**Table 1 animals-13-00978-t001:** Behavioural parameters registered during the dominance test performed in Senegalese sole. Behaviour, acronym and description of each behaviour are indicated.

Behaviour	Acronym	Description
**Position before feeding**	POSITB	Index per fish per day = ((position 1 x “y”) + (position 2 x “y”) + (position 3 x “y”) + (position 4 x “y”) + (position 5 x “y”) + (position 6 x “y”))/12 (nº of fish)
		“y” = Frequency (events) of each position during the 30 min before feeding (9:30–10 AM) (Without and With sand)
**Position after feeding**	POSITA	Index per fish per day = ((position 1 x “y”) + (position 2 x “y”) + (position 3 x “y”) + (position 4 x “y”) + (position 5 x “y”) + (position 6 x “y”))/12 (nº of fishes)
		“y” = Frequency (events) of each position during the 30 min after feeding (10–10:30 am) (Without sand) “y” = Frequency (events) of each position from the beginning of food delivery until the last pellet was eaten (10–12:00 pm) (With sand)
**Feeding order**	FO	Fish that ate first was 1, second to eat was 2, etc (rank according to feeding)
**Rest the head**	RTH	Percentage of the number of times (events) that a fish rests its head on another fish before and after the feeding period (Without sand: 60 min; with sand: 150 min).

**Table 2 animals-13-00978-t002:** Primers used for exploring transcript abundance in the brain of Senegalese sole (*Solea senegalensis*) related to dominance category and presence of an enriched environment. Gene, gene name, amplicon size, primer sequence and biological function are indicated.

Gene	Gene Name	Amplicon Size	Primers (5′ 3′)	Biological Function
Ubiquitin	ubq	89	F-AAAATTCCCCAATCAATCTCCT R-CTTCACAAAGATCTGCATCTTGA	Housekeeping [28]
Beta-actin	b-act	90	F-GCCTTTGCCGATCCGC R-GCCGTAGCCGTTGTCG	Housekeeping [28]
C-FOS	c-fos	175	F-CTGGAGTTCATTCTGGCTGC R-TTGAGGTGAATGTTGGCTGC	Neuroplasticity, neurogenesis, and brain activation [25]
Nuclear Receptor Subfamily 4, Group A, Member 2	nr4a2	187	F-TCTCCCGAGTTTCAGCACTT R-CCCAGAGTGAGCCATCATTT	Differentiation of dopamine neurons [26]
Neurogenic differentiation factor 2	nrd2	396	F-TTATCAGTGTGCGCGTCTGT R-TTCAGTTCGTCGTACACGGG	Neuroplasticity, neurogenesis, and brain activation [25]
Brain-derived neurotrophic factor	bdnf	154	F-ACTCGTTTGAAACATCCGGC R-CAGACAGGGTGAGTGGAGAA	Neuroplasticity (involved in changes in synaptic plasticity), neurogenesis, and brain activation [24]
Heat shock protein 90, alpha (cytosolic) class A	hsp90aa	105	F-GACCAAGCCTATCTGGACCCGCAAC R-TTGACAGCCAGGTGGTCCTCCCAGT	Cellular stress, cellular protection, and cellular homeostasis [27]
Heat shock protein 70	hsp70	119	F-AGCCACCGTGTCGCCGACCT R-CGACCTCCTCAATATTTGGGCCAGCA	Cellular stress and cellular protection [27]
Glyceraldehyde-3-phosphate dehydrogenases 2	gapdh-2	107	F-AGCCACCGTGTCGCCGACCT R-AAAAGAGGAGATGGTGGGGGGTGGT	Metabolic pathway, membrane fusion, phosphotransferase activity, nuclear RNA export, DNA replication and repair, apoptosis, age-related neurodegenerative disease, and viral pathogenesis [28]
Nuclear progestin receptor	pgr	120	F-TGTCTGACCACCTTCATCCA R-TCCAGTCACAGGACGTCTCC	Gene expression in reproductive tissues, progestin-induced signalling pathways to regulate spermatogenesis and testicular differentiation and spermatogenesis [25]
Follicle-stimulating hormone receptor α	fshra	60	F-GGACCCAAACTACATCCATGAAC R-CAGTCCCCGTTACAGATCACCTGTCT	Supports steroidogenesis, folliculogenesis, and ovulation and stimulates spermatogenesis [31,32]

**Table 3 animals-13-00978-t003:** Growth parameters in Senegalese sole of dominants (*n* = 6), intermediates (*n* = 12) and subordinates (*n* = 6) without sand and dominants (*n* = 6), intermediates (*n* = 12) and subordinates (*n* = 6) maintained with sand. Data are shown as mean ± SD.

Treatment	Social Category	Initial Weight (g)	Final Weight (g)	The Mean of Weight Gained in 2 Months (g)
Without sand	Dominant	205.33 ± 13.44	342.51 ± 38.70	137.17 ± 26.97
	Intermediate	203.92 ± 17.39	319.58 ± 34.94	115.66 ± 32.87
	Subordinate	204.17 ± 11.96	308.83 ± 58.44	104.67 ± 53.43
With sand	Dominant	197.83 ± 15.69	314.51 ± 44.14	116.68 ± 29.19
	Intermediate	204.29 ± 16.57	309.19 ± 34.19	104.91 ± 32.93
	Subordinate	205.67 ± 18.41	308.83 ± 58.44	111.63 ± 50.38

## Data Availability

Raw data from behavioural analysis and qPCR expression are available in figshare (https://doi.org/10.6084/m9.figshare.16910380.v1).

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
