# Peer review of "The Use of Sand Substrate Modulates Dominance Behaviour and Brain Gene Expression in a Flatfish Species"

_animals, 2023, doi:10.3390/ani13060978_

Round 1
Reviewer 1 Report
Several studies have been conducted in recent years in an attempt to address the reproductive problem that Senegalese sole males exhibit in captivity, including studies with the application of hormone therapies to behavioral trials. Nonetheless, reproduction-related problems still remain. The reproductive problem of males of culture origin has led to the need for wild animals in production systems for the obtention of fertilized spawns, which is far from sustainable. Studies focused on the domestication process of this species are welcome and of importance for the European aquaculture sector. Almeida et al. examined the effects of an enriched rearing environment on Senegalese sole dominance behavior and at the brain gene expression level. Results are promising in the enhancement of the welfare and maturation of this species in captivity. This manuscript is acceptable for publication but there are a few minor corrections and suggestions to be addressed:
Line 8. Indicate how the actual rearing conditions of juveniles are. An example could be “intensive rearing conditions in fiberglass or plastic tanks”. Since the environmental enrichment is the addition of sand, the reader needs to make a picture of how the non-enriched environment is.
Line 17 and so on (lines 35, 364, 366, 464, 486, 492). Please, indicate genes in italics.
Line 23. Correct to “reduces”.
Line 25. Indicate that the reproductive dysfunctions “do not allow” to close the life cycle in captivity. Otherwise, the sentence is not well formulated.
Line 68. Correct to “has also been”.
Line 90. Correct to “individuals”.
Line 96. Add “studies in other fish species”.
Line 106. Correct to “has been observed to be differentially expressed”.
Line 115. Please, indicate if juveniles are of culture origin.
Line 123. Did the authors mean shade nets? Rewrite to “to avoid direct light contact due to Senegalese sole sensitivity to light”.
Line 125. Put “ad libitum” in italics.
Line 126. Change to “on a daily basis” since it is a bit awkward to say 7 days per week. Perhaps the authors meant Lda instead of Ida. Please, indicate the country where SPAROS is.
Line 148. Correct to “focused”.
Line 155. Indicate why the test was performed at 10 am. Is it sole’s peak of activity? Is it a diurnal species?
Line 173. How the fish that were placed between the two zones were classified? Were these zones painted in the tanks? How did the examiner establish these different zones?
Table 1. In the description of POSITA “y” the authors should indicate that it was = Frequency (events) of each position from the beginning of food delivery until the last pellet was eaten, as indicated in the text. In the case of the group without sand, it corresponded to 30 min, but in the group with sand did not correspond to 30 min. Correct this, please.
Line 301. Simplify to “with and without sand”. Correct to “However, there was”.
Figure 4 caption. Indicate that the asterisk denotes significant differences between treatments “within the same category of dominance”.
Line 450. Add “, and may alter”
Table 2. The reference for Chauvigné et al 2012 is missing from the list of references.
Author Response
Please, see the attachment.

Reviewer 2 Report
This is an important and well-designed study to evaluate the effects of environmental enrichment in the sole (Solea senegalensis ) on dominance behaviour and gene expression in the brain. Ultimately, the authors propose to assess the impact of environmental enrichment on the welfare of the species studied.
The manuscript contains some points that deserve special mention, such as the good use of figures (especially Figures 1 and 2) to illustrate the methodology used and the effort to describe in detail the results obtained, especially concerning the dominance analysis.
On the other hand, some points still require additional efforts from the authors to make their work ready for publication. I will present some observations on the authors' analyses and considerations in the following.
Essay
There are many (minor) typos. I have converted the article to Word (docx) to point out these errors more easily. I emphasise that my critique on this topic was superficial and only meant to point out that the wording needs improvement. I also suggest that the authors use either American or British English, as currently, both forms appear equally in the text.
Methodology
I understand that the experimental methodology should be better described in some points. For example:
What criterion or reference was used to define the amount of sand (less than 1 cm) added in each treatment?
Why are fish fed only once daily and at 2% of their biomass? What reference was used to support this method? Does feeding only once a day affect the welfare of the fish?
I have the same doubts about the density/biomass of the animals per tank. In Figure 1, you will notice that the fish are very close to the bottom of the tanks, which does not fit a study designed to evaluate the welfare of farmed fish.
Regarding tags, I understand that the description is inadequate, and I could not understand what exactly was done. How were the tags applied, and when did this occur? The text does not address that the welfare of the animals is affected by the use of markers.
Results
I think the description of the results in terms of dominance is adequate, and the report of the brain gene expression analysis results is somewhat confusing. Sometimes I feel that the authors go beyond what is acceptable in describing the results as if they are forcing the data to confirm their hypotheses. For example, when they write:
Instead of this fact, transcript nr4a2 (P = 0.0945; Fig 6A) showed to be slightly lower without being significant in dominant fish than subordinates in animals reared without sand.
However, the transcript c-fos (P = 0.064; Fig. 6C) showed to be slightly higher expressed without being significant by dominant fish maintained in the sand.
Table 3 is incomplete, in my opinion. The authors describe in Figure 2 that the duration of the acclimation period was 2 months, that the acclimation period to the sand lasted 5 days, 3 more days for acclimation to the presence of the feeding tubes, 2 more days of fasting, and finally, 4 more days of the actual experiment. That is a total of 2.5 months. However, the differences between treatments would not be noticeable until the last 14 days. Therefore, I understand that more biometric data would have been needed before this previous phase to interpret the results in Table 3. It makes no sense to compare weight gain in 2 months when the experiment lasted 2.5 months, and the observed differences were not apparent until the last 14 days.
I suggest carefully revising item 3.2 to make it more objective and meaningful.
Discussion
It is clear to me that, as in the last part of the description of the results, there are problems and a need for adjustment in the discussion. It starts with the first paragraph, which is strange and has become a kind of summary of the results. I suggest deleting it because this paragraph is unnecessary if the results are described objectively and clearly.
Some ideas need to be adjusted. What the authors say actually needs to be rigorous. For example, the authors use the terms "behavioural pattern" and "events" in the sentence: "The results suggest that dominance expression was decreased in the presence of sand. Indeed, the behavioural pattern for all the parameters registered of fish from both treatments was similar, however, the frequency of events was lower in fish reared in sand than in sole maintained without sand". The precise definition of each of these terms needs further discussion in the text.
In other places, it appears that the authors are once again trying to force the data to support their hypotheses when the problems are more complex. For example, in the text: "This could be explained by the fact that animals kept in the sand showed a delay in terms of feeding behaviour just starting to eat 30 minutes after the food was provided, being buried in the sand, and remaining quiet. Therefore, the substrate might be considered a key factor for sole welfare in aquaculture". Of course, this is about the welfare of the farmed fish. However, it does not seem to be a good indicator of welfare if the fish do not start eating until 30 minutes after feeding since they have lost a large proportion of their nutrients through hydrolysis. This point has been neglected in the discussion, and in my opinion, it is an essential factor in the welfare of the fish.
On the other hand, the authors themselves admit: "The present study investigated some of the target genes involved in several functions (cognitive capacity, stress modulation, basic metabolism and maturation). In our study none of the transcripts analysed (nr4a2, c-fos, nrd2, bdnf, hsp90aa, hsp70, gapdh-2, pgr, fshra) were significantly differentially expressed between dominant and subordinate sole in both environments, with and without sand". However, further down they write, "In our study it was possible to observe that individuals reared in sand showed higher maturity, since dominant individuals from sand presented higher expression of fshra than dominant individuals from tanks without sand". I point out again: If the results were not different, one could not assume differences between treatments. The authors must describe their effects more clearly and stick to them better when discussing the results.
The authors point out that: " In the present study, a small quantity of sand was used to facilitate the observation of the fish and to perform the maintenance routines. No mortalities or signs of sickness were observed during the period of the experiment and the parameters of the water were registered every day without noticing changes between treatments". However, they did not consider that this sand remained in the tanks for only 14 days, and this time would be sufficient to overcome the adverse effects of the long-term presence of sand on the welfare of the farmed fish. Again, I understand that no.
The authors take a relatively large amount of space to discuss the effects of sand on gonadal maturation in fish, but they forgot to consider that the differences between treatments only occurred during 14 days (or less). They also did not address the size of the animals at first maturation and did not perform a gonadal analysis at the end of the experiment. Under these conditions, is it possible to sustain the discussion and subsequent conclusions about sand's effects on organisms' gonadal maturation? I cannot understand this. But if the authors want to go this way, they must go much deeper in analysing their own results.
Conclusion
The conclusions look more like a summary of the work done than conclusions, and I propose to mention only the findings in this article.

Author Response
Please, see the attachment file

Round 2
Reviewer 2 Report
Dear authors,
I congratulate you on the work you have done and hope that I have been able to play my part in it
Author Response
The authors thank the suggestions made by the reviewer who has helped us to improve to manuscript considerably to be considered for publication.